# Immunotherapy in Hematologic Malignancies: Emerging Therapies and Novel Approaches

**DOI:** 10.3390/ijms21218000

**Published:** 2020-10-27

**Authors:** Ji-Yoon Noh, Huiyun Seo, Jungwoon Lee, Haiyoung Jung

**Affiliations:** 1Immunotherapy Research Center, Korea Research Institute of Bioscience and Biotechnology (KRIBB), 125 Gwahak-ro, Yuseong-gu, Daejeon 34141, Korea; nohj16@kribb.re.kr; 2Center for Genome Engineering, Institute for Basic Science (IBS), 55 Expo-ro, Yuseong-gu, Daejeon 34126, Korea; hyseo@ibs.re.kr; 3Environmental Disease Research Center, Korea Research Institute of Bioscience and Biotechnology (KRIBB), Yuseong-gu, Daejeon 34141, Korea; 4Department of Functional Genomics, Korea University of Science and Technology (UST), 113 Gwahak-ro, Yuseong-gu, Daejeon 34113, Korea

**Keywords:** hematologic malignancy, immune checkpoint, chimeric antigen receptor, lymphocyte, antibody–drug conjugate

## Abstract

Immunotherapy is extensively investigated for almost all types of hematologic tumors, from preleukemic to relapse/refractory malignancies. Due to the emergence of technologies for target cell characterization, antibody design and manufacturing, as well as genome editing, immunotherapies including gene and cell therapies are becoming increasingly elaborate and diversified. Understanding the tumor immune microenvironment of the target disease is critical, as is reducing toxicity. Although there have been many successes and newly FDA-approved immunotherapies for hematologic malignancies, we have learned that insufficient efficacy due to disease relapse following treatment is one of the key obstacles for developing successful therapeutic regimens. Thus, combination therapies are also being explored. In this review, immunotherapies for each type of hematologic malignancy will be introduced, and novel targets that are under investigation will be described.

## 1. Introduction

The theory of tumor immunosurveillance and the potential of immunotherapy regimens were first suggested more than 50 years ago, and, as of recently, the idea is receiving great attention due to the beneficial responses observed in certain groups of cancer patients receiving immunotherapy. Particularly, immune checkpoint inhibitors (ICIs), such as nivolumab, cause tumor remission in nonsmall cell lung cancer (NSCLC) and metastatic melanoma, whereas chimeric antigen receptor (CAR)-T cell therapy, such as tisagenlecleucel, has had great success in B cell acute lymphoblastic leukemia (B-ALL) [1]. Regardless of the success of immunotherapy in oncology, not all patients benefit from these recently developed regimens. This is mainly due to differences in the availability and immunogenicity of tumor antigens, exhaustion of cytotoxic lymphocytes, and a variety of tumor escape mechanisms. Therefore, it is important to thoroughly understand immunotherapy’s mechanism of action, the tumor microenvironment, tumor immune signatures, as well as cancer cell-intrinsic genetic and epigenetic aberrations [2].

Hematologic malignancies can be driven by genetic or epigenetic changes within hematopoietic cells as well as changes in the stromal niche, including hypoxia, angiogenesis, and inflammation. Malignant hematopoietic cells can resist chemotherapy, facilitate immune evasion, and survive in the remodeled niche [3]. The goal of the current review is to summarize advances in immunotherapy approaches for the treatment of hematologic malignancies.

## 2. Principles and Types of Tumor Immunotherapy

### 2.1. Cytokines

Cytokines, including interleukin-2 (IL-2) and IL-15, enhance the proliferation and activation of CD8^+^ T cells and natural killer (NK) cells [4]. In fact, interferon-alpha (IFN-α) was approved for the treatment of hairy cell leukemia, while IL-2 was introduced for the treatment of advanced melanoma in 1986 and 1998, respectively [5]. IL-2 was also infused after haploidentical NK cells were administered for treating high-risk acute myeloid leukemia [6]. An agonist of the IL-2 receptor-beta (CD122), bempegaldesleukin (NKTR-214), has been shown to increase the number and activity of CD8^+^ T cells without affecting Foxp3^+^ regulatory T (Treg) cell numbers [7]. Cytokines may also be used in combination with ICIs or CAR-T, and CAR-NK cells. For instance, methods for autologous stimulation via IL-2 have been developed for CAR constructs or through gene editing in NK-92 cells. In addition, dendritic cell (DC)-derived IL-12 plays a critical role in immune checkpoint blockade (ICB) therapy [8]. Overall, cytokines boost immune system responses, including antitumor immunity.

### 2.2. Immune Checkpoint Blockade (ICB)

T cells employ two mechanisms for the killing of target tumor cells: one is through antigen-specific signaling via T cell receptors (TCRs), and the other is through antigen-nonspecific signals. The latter is associated with costimulatory receptors (e.g., CD28) or coinhibitory receptors (e.g., CTLA-4 and PD-1). Cytotoxic T cells and NK cells are suppressed upon the engagement of coinhibitory receptors, allowing for the immune escape of target tumor cells [9]. Thus, blocking inhibitory checkpoints with ICIs can harness immune cells to effectively attack tumor cells. It has been demonstrated that ICIs can indeed successfully extend survival for months or years in patients who would otherwise survive for less than a year on other recommended treatment regimens [10]. Since 2011, seven ICB therapeutics, including ipilimumab, nivolumab, and pembrolizumab, have received FDA approval for the treatment of metastatic melanoma, advanced NSCLC, Hodgkin lymphoma, and other malignancies. [11].

CTLA-4 is expressed on activated T cells, including Treg cells. Upon T cell receptor (TCR) activation, CTLA-4 is upregulated and interacts with CD80 on antigen-presenting cells (APCs), resulting in T cell-intrinsic suppression [12]. For CD80 binding, CTLA-4 competes with T cell costimulatory receptor CD28, indicative of the regulatory role of CTLA-4 in T cell activation [13]. In addition, CTLA-4 engagement in Treg cells can induce the release of an immunosuppressive mediator indoleamine 2,3-dioxygenase (IDO), suggesting that CTLA-4 plays a crucial role in Treg-mediated immune tolerance [14,15]. Recently, CTLA-4 recycling is considered to be an important factor for preventing immune-related adverse events (irAEs) during the use of anti-CTLA-4 monoclonal antibodies (mAbs) in cancer. CTLA-4 is colocalized with lipopolysaccharide-responsive and beige-like anchor (LRBA) protein in endosomes, where it can be recycled or directed to the lysosome for degradation [16]. Interestingly, the irAE-prone anti-CTLA-4 mAbs (e.g., ipilimumab) directed lysosomal degradation of surface CTLA-4 by preventing its interaction with LRBA. Recently, a novel anti-CTLA-4 mAb has been developed. It was engineered to dissociate from CTLA-4 in endosomal vesicles due to its pH level, leading to the improvement of irAE upon using the anti-CTLA-4 mAb. It demonstrates that the targeting of CTLA-4 needs to be tightly controlled [17].

PD-1, upon engagement of its ligands PD-L1 or PD-L2, recruits tyrosine phosphatase SHP2 and inhibits the T-cell-receptor-mediated intracellular activation signaling cascade [18]. CD28 is known to be the most sensitive target of PD-1-SHP2. PD-1 also represses the expression of genes transcribed following strong TCR signals, such as those encoding cytokines and effector molecules [19]. The PD-L1/CD80 *cis*-heterodimer can be found on APCs, affecting the *trans*-interaction with PD-1, CTLA-4, or CD28, suggestive of a competitive interaction. Detailed investigations on the use of ICIs in hematologic malignancies will be further discussed in the next section.

### 2.3. Antibody–Drug Conjugates (ADCs)

In general, chemotherapy employs a cytotoxic compound that affects the whole body. Thus, targeted drug delivery systems have long been desired and optimized to improve the efficacy of chemotherapy agents by minimizing adverse effects in healthy tissue. The introduction of monoclonal antibodies allowed researchers to employ their specificity as a mechanism for targeted drug delivery. Hence, antibody–drug conjugates (ADCs) were developed by chemical conjugation of a cytotoxic agent to a tumor-targeting antibody [20]. The antibody recognizes tumor cell-specific or cell-enriched antigens and delivers a highly potent DNA-binding agent, attached to the antibody via a cleavable linker [21].

In 2000, the first ADC, gemtuzumab ozogamicin, was approved for treating acute myeloid leukemia [22]. It comprises an anti-CD33 antibody conjugated to calicheamicin, a potent DNA-targeting agent. It was recently reapproved by the FDA, after being withdrawn in 2010. Tagraxofusp, a fusion protein consisting of IL-3 and diphtheria toxin, was FDA-approved for blastic plasmacytoid dendritic cell neoplasm (BPDCN) in 2018. BPDCN is a rare and clinically challenging hematologic malignancy; however, a breakthrough has been made by detecting CD123, a surface receptor for IL-3, which was overexpressed in the tumor cell. The overall response rate was 90% with tagraxofusp and further studies to improve outcomes will be continued [23].

ADCs have also been successful in solid tumors; for example, ado-trastuzumab emtansine for the treatment of HER2-positive breast cancer [24]. More clinical trials are underway for ADCs against solid tumors. It has been suggested that the level of target antigen expression in tumor cells and the selection of patients based on diagnostic test results are essential for ADC treatment efficacy.

### 2.4. Bispecific T/NK Cell Engagers (BiTEs/BiKE)

Bispecific antibodies may recognize both T cells or NK cells and tumor-associated antigens (TAA), thus directing immune cells to target cancer cells more effectively. Typically, bispecific T cell engagers (BiTEs) have been developed against CD33/CD3 and CD123/CD3 for the treatment of hematologic malignancies [25,26]. Similarly, bi- and trispecific NK cell engagers (BiKE and TriKE) MLLhave also been developed, linking activating NK receptors (e.g., CD16) to TAAs. The promising results with this strategy led to the achievement of the FDA approval of blinatumomab for treating B-ALL [27]. Blinatumomab is a BiTE consisting of variable regions of anti-CD3 and anti-CD19 and has brought a significant treatment advancement for patients with relapsed or refractory and/or minimal residual disease-positive B cell ALL [28,29]. Nonetheless, an obstacle has been suggested for employing BiTEs where the therapeutic efficacy could be impeded due to T cell exhaustion or anergy [30]. TAA-positive tumor cells often upregulate the surface level of PD-L1, and the activity of T cells recruited to PD-L1-high tumor cells would be compromised via the PD-1 signaling cascade. Thus, combination therapy with PD-L1/PD-1 blockade and BiTE antibodies has been investigated, and it has resulted in enhanced T cell activity, as indicated by higher IFN-gamma production [31].

### 2.5. Chimeric Antigen Receptor (CAR)-T, CAR-NK Cells

CAR-T cell therapy is an adoptive cellular immunotherapy using genetically modified lymphocytes. Prior to infusion into the patient, T cells are collected from the patient, expanded in a bioreactor, and modified to express a specific CAR [32]. The CAR consists of an extracellular domain, a single-chain variable fragment (scFv) that recognizes the tumor antigen, a transmembrane domain, and an intracellular T cell activation domain, most often CD3z. In addition to CD3z, intracellular domains of T cell receptor costimulatory molecules, such as CD28 and/or 4-1BB, are necessary to ensure the persistence and efficacy of CAR-T cells, and the incorporation of these domains led to the development of second- and third-generation CARs [33,34]. Subsequently, fourth-generation CARs called T cells redirected for antigen-unrestricted cytokine-initiated killing (TRUCKs) have also been developed. These can further induce cytokine production or apoptotic protein expression upon activation [35].

The FDA approved three CAR-T cell therapies in 2017 and 2020, namely tisagenlecleucel, axicabtagene ciloleucel, and brexucabtagene autoleucel [36,37]. They express an scFv derived from the mouse monoclonal antibody FMC63, which specifically recognizes human CD19. With regard to the costimulatory intracellular domain, axicabtagene ciloleucel and brexucabtagene autoleucel are composed of CD28, whereas tisagenlecleucel contains 4-1BB. Although the scFv is a critical factor that determines the target of CAR-T cell therapy, the effect of other domains, including the linker [38], hinge [39], transmembrane, and intracellular/costimulatory domains [40,41], should be thoroughly investigated, as it could modulate efficacy and associated adverse events.

Although CAR-T cell therapies are innovative and have been successful in certain patients, about 70% of patients who receive CAR-T cells fail to respond or relapse after therapy [42,43]. Therefore, extensive efforts have been made to enhance CAR-T cell activity through the development of next-generation CAR constructs and combination therapy with ICB or other CARs. For instance, various CAR-T cells have been introduced that can generate immunostimulatory ligands and cytokines, such as the CD40 ligand [44], Fms-related tyrosine kinase 3 (FLT3) ligand [45], IL-12 [46], and IL-18 [47], upon engagement. Immune checkpoint receptor PD-1 can also be upregulated in activated CAR-T cells, and, recently, various studies have reported that suppression of PD-1 in parallel to CAR-T therapy could be beneficial. Further, genome engineering techniques such as CRISPR/Cas9 [48] and TALEN [49] were used to generate PD-1-deficient CAR-T cells. CAR-T cells were also modified to secrete PD-1-blocking scFv, which improved the antitumor response. A dual CAR-T targeting both CD19 and CD20 by transduction of CAR containing tandem scFv to T cells has also been investigated, which results in an elicited antitumor response [50]. In addition, the limited autologous T cell expansion could be overcome through the development of TCR alpha removed CAR-T cells or allogeneic CAR-NK cells, which are being investigated [51].

CARs can be introduced into NK cells, generating CAR-NK cells, which are generally thought to be safer than CAR-T cells [52]. Recently, promising results from a clinical trial using anti-CD19 CAR-NK cells were reported [53]. CAR-NK cells were infused to patients with relapsed or refractory CD19-positive hematologic malignancies, and complete remission (CR) was seen in 64%. The CAR construct possesses IL-15, and the authors suggested that this might have played an important role in the persistence of CAR-NK cells with functional activity in vivo for several months. Notably, the NK cells were derived from HLA-mismatched cord blood, suggesting the advantages of employing NK cells over T cells. More importantly, the recipients did not represent significant adverse events by CAR-NK cells, such as cytokine release syndrome.

## 3. Hematologic Malignancies and Immunotherapy

Hematologic malignancies develop as a result of genetic and epigenetic changes that accumulate in hematopoietic cells. There are several different types of hematologic malignancies with different etiologies. It can be categorized by the affected hematopoietic cell type as well as the place where the tumor occurs. In this section, the pathophysiology of hematologic malignancies and the relevant immunotherapies will be summarized.

### 3.1. Acute Myeloid Leukemia (AML)

Acute myeloid leukemia (AML) represents the accumulation of immature myeloid progenitor cells in the bone marrow and peripheral blood. It is reported that AML occurs at all ages, but the median age of diagnosis is >60 years, indicating that incidence rates are higher in the elderly [54]. The onset of AML is typically preceded by somatic mutations in HSPCs. The mutated HSPCs proliferate extensively, a process known as clonal hematopoiesis, and therefore, the mutations can be accumulated during aging [55]. Genetic mutations that are commonly observed in AML include *KIT, RUNX1*, *CEBPA*, *TP53*, *NRAS*, *PTPN11*, *NF1*, *GATA2*, *FLT3*, *and NPM1*. In addition, the epigenetic regulators mutated in AML include *DNMT3A*, *TET2*, *IDH1*/*2*, *MLL*, and *PRMT*, which are mostly associated with DNA methylation, one of the major epigenetic mechanisms controlling hematopoietic differentiation [56,57]. Leukemia stem cells (LSCs), transformed cells that give rise to AML blasts while retaining their self-renewal capacity, have been studied for the last two decades in AML, and therapeutic approaches have focused on targeting LSCs [58]. Once LSCs acquire the self-renewal ability through additional driver mutations, AML is likely to occur. LSCs could also involve in post-treatment relapse, of which mechanism is well-known as HLA loss in AML [59].

Despite extensive studies and the well-defined mechanism of AML, treatment methods have not changed much in the past 30 years, and, more importantly, AML is successfully cured in only 35–40% of patients <60 years of age and 5–15% of patients >60 years of age [60]. Of note, AML was the first malignancy in which durable remissions were achieved by allogeneic hematopoietic stem cell transplantation (HSCT), the most potent antileukemic, and an immunotherapeutic approach [61]. It is believed that its curative effect is based on the graft-versus-leukemia effect of allogeneic T cells on AML cells. However, due to toxicity, allogeneic HSCT is often not an option, particularly for older patients. Thus, immunotherapies with a precise mode of action and less toxicity are required. Therapeutic targets for AML include the cell cycle, metabolism, epigenetic molecules, tumor cell surface antigens, and immune checkpoints. The current review focuses on immunotherapies.

#### Immunotherapy for AML

Developing effective immunotherapy for AML has been challenging. AML cells are heterogeneous, which contributes to the inability of the immune system to recognize tumor-specific markers [62,63]. Indeed, AML cells can develop immune escape mechanisms effectively avoiding death, for instance, by suppressing NK cells or reducing the expression of certain surface receptors. AML cells also increase expression of inhibitory immune checkpoints including PD-L1, PD-L2 [64], CD47 [65], and CD70 [66]. Therefore, many trials on immunotherapies and/or combination therapies are being conducted with promising results. Anti-PD1 or anti-CTLA4 therapy following disease remission after chemotherapy has been tested for eliminating measurable residual disease (MRD), demonstrating strong T cell responses against AML [67]. The interaction between CD70 and CD27 is one of the immune escape mechanisms along with the increased frequency of Treg cells or enhanced clonal expansion of AML cells via TRAF2- and TNIK-mediated canonical Wnt pathway activation [68]. A CD27-targeting mAb (varlilumab) effectively eliminated CD27-expressing lymphoma and leukemia [69]. An anti-CD70 mAb (cusatuzumab) has also been developed and proved to be a promising therapeutic approach in preclinical models of AML [70].

Since the first approval of ADC, gemtuzumab ozogamicin, comprising an anti-CD33 antibody conjugated to calicheamicin [22], approximately 80 ADCs have been developed and assessed in nearly 600 clinical trials [71]. The first ADC was approved for AML and vadastuximab talirine, also known as SGN-CD33A, in combination with hypomethylating agents or pyrrolobenzodiazepine, was evaluated in clinical studies for newly diagnosed or relapsed AML as well as for newly diagnosed MDS. SGN-CD123A is also in clinical development for treating AML [72].

A BiTE, anti-FLT3/CD3 (AMG-427) is currently being evaluated in a clinical study (NCT03541369, phase I) [73]. Anti-CD33/CD3 (AMG330) [74] and anti-CD123/CD3 [75] have also been suggested as treatments for AML. *FLT3* mutation is the most common mutation observed in AML (about 30%) and induces the ligand-independent, constitutive activation of the receptor tyrosine kinase, enhancing cell survival. Similarly to FLT3 inhibitors (e.g., gilteritinib), antibodies against FLT3 are also effective for reducing cell growth. Furthermore, FLT-3-targeting BiTEs are able to recruit cytotoxic T cells to destroy tumor cells. CD123 is an IL-3 receptor subunit and is considered an LSC marker. CD123 is expressed on CD34^+^CD38^-^ AML cells, and these cells with CD34^+^CD38^-^CD123^+^ were able to engraft in immunodeficient mice [75]. Thus, CD123 represents a promising target molecule for the detection of AML cells without affecting the healthy bone marrow cells. The overexpression of CD123 is associated with the constitutive phosphorylation of STAT5, accelerated cell proliferation, and reduced apoptosis [76]. Other antibodies against C-type lectin domain family 12 member A (CLL-1) [77], mainly expressed in AML LSC specifically, CD47 [78], and IL-1 receptor accessory protein (IL1RAP), which stimulates oncogenic activity in AML through activation of the innate immune signaling pathway [79], have also been suggested as therapeutic candidates.

The antigens described above can be exploited as targets of CAR-T cells. CD33 CAR-T therapy was tested in combination with autologous CD33-knockout bone marrow transplantation using a gene-editing tool, such as CRISPR/Cas9 [80]. Since CD33 is expressed in normal HSCs but has no relevant function, this approach was feasible. CD123 CAR-T cells are also in clinical development. CD117 CAR-T cells were recently reported to efficiently eliminate AML blasts as well as CD117^+^ healthy HSCs in the AML model [81]. CD117, a receptor tyrosine kinase to which stem cell factor binds, is expressed on hematopoietic precursors; however, it may remain overexpressed following malignant transformation in HSCs [82,83]. In a recent study, the CAR was modified to comprise an anti-CLL-1 scFv linked to an anti-CD33 scFv via a self-cleaving P2A peptide, resulting in the expression of both functional CARs on the T cell surface [84]. Another dual CAR-T cell therapy employing anti-CD123/CLL-1 is currently being evaluated in a clinical trial (NCT03631576). Similarly to CAR-T cell therapy, adoptive cell transfer is a promising treatment method for the stimulation of patients’ immunity. In addition, harnessing NK cells for adoptive cell transfer is feasible, as alloreactive NK cells from the donor can suppress leukemic cells and LSCs, as shown in a patient-derived xenograft (PDX) animal model with PARP inhibitor cotreatment [85]. Moreover, personalized dendritic cell vaccines, namely DC/AML fusion cells, can be infused into patients, resulting in AML remission [86]. Vaccination and/or transfer of antigen-specific CD8^+^ T cells with WT1 extended overall survival in AML [87,88]. Taken together, with the technical advances in gene editing and ex vivo expansion of human immune cells, adoptive immune cell transfer methods are quickly improving and contributing to personalized medicine.

### 3.2. Myeloproliferative Neoplasm (MPN)

Myeloproliferative neoplasms (MPN) are blood cancers that occur in the bone marrow. In MPN, one or more types of blood cells are produced highly, resulting in blood thickening or dysregulation of bone marrow. It includes seven types: chronic myeloid leukemia (CML), chronic neutrophilic leukemia, chronic eosinophilic leukemia-not otherwise specified, primary myelofibrosis (PMF), polycythemia vera (PV), essential thrombocythemia (ET), and MPN, unclassifiable (MPN-U) [89]. CML is normally diagnosed with the invariable presence of the *BCR-ABL1* mutation (Philadelphia chromosome mutation). On the other hand, among the *BCR-ABL1*-negative MPNs, PV, ET, and PMF show *JAK2*/*CALR*/*MPL* mutation. CML is relatively well controlled by imatinib, a kinase inhibitor. Nevertheless, *JAK2*-mutated clones are sometimes detected in CML patients following treatment [90].

Approximately 50% of PMF cases with the highest mortality among *BCR-ABL1*-negative MPNs harbor mutations in *ASXL1*, *EZH2*, *TET2*, *IDH1*/*2*, or *SF3B1*, representing clonal hematopoiesis [91]. PMFs are diagnosed with elevated counts of white blood cells as well as fibrosis in bone marrow and spleen, while ETs and PVs are characterized by an overproduction of platelets and red blood cells, respectively [92]. These Philadelphia chromosome-negative MPNs can be transformed to AML approximately in 11–20%, 1–5%, and 4–7% of PMF, ET, and PV cases, respectively, and it is considered one of the major complications of MPN [93].

#### Immunotherapy for MPN

Among disease types of MPN, the Philadelphia chromosome-negative MPNs, PV, ET, and PMF are considered chronic and inflammation-related diseases, which implicates the dysregulation of the immune system. Thus, immunotherapies are tested and conducted to treat these types of MPN [94]. IFN-alpha has been used for the management of myelofibrosis in MPNs for over 30 years. It has recently been recommended as an alternative to ruxolitinib for the treatment of low-risk MPNs by the National Comprehensive Cancer Network [95]. IFN-alpha has antiproliferative, proapoptotic, and immunoregulatory effects, which could modulate disease symptoms related to the aberrant immature megakaryopoiesis and granulocytosis observed in MPNs [96]. In line with this notion, a long-acting novel IFN-alpha, ropeginterferon-alpha 2b, has been approved for treating PV patients in Europe.

Interestingly, identifying neoantigens in MPNs for monoclonal antibody therapies or utilizing ICIs may lead to better results in MPNs in comparison to MDS and AML. It has been reported that JAK2 mutations can trigger the upregulation of PD-L1 on myeloid cells and PD-1 ICB improved disease symptoms in a human MPN xenograft murine model [97]. Notably, neoantigens can be also identified in association with mutations of *JAK2V617F* (>50%), *MPL* (3–5%), and *CALR* (20–30% in ET and PMF) in MPN, which would elicit tumor-specific T cell responses [98,99]. However, treatment with IFN-alpha or ICB has limited effects on patients who are refractory to JAK2 inhibitors, suggesting that there is an unmet need for the treatment of MPN through immunotherapies [100].

### 3.3. Hodgkin Lymphoma (HL)

Hodgkin lymphoma (HL) accounts for approximately 10% of lymphoma cases. It is developed in the lymphatic system and occurs mostly sporadically. It can also be associated with the Epstein-Barr virus (EBV) or HIV/AIDS and originates from the lymph node [101]. Individuals with classical HLs have malignant Reed–Sternberg (RS) cells in the lymph node. Importantly, HL could involve abnormal amplification of chromosome 9p24.1, a locus containing JAK2, PD-L1, and PD-L2 [102]. The abnormality not only directly induces PD-L1 and PD-L2 expression but also increases *PD-L1* transcription through gene dose-dependent JAK-STAT activity. Furthermore, EBV infection could lead to the upregulation of PD-L1 in EBV-positive malignant cells [103]. The tumor microenvironment of classical HL consists of abundant and ineffective immune cells and rare malignant cells, in contrast to that of non-HL [104].

#### Immunotherapy for HL

The above-described features of HL have led to the investigation of the efficacy of PD-1 ICIs in HL. In a clinical trial, the rate of progression-free survival at 24 weeks was 86%. Nivolumab substantially improved disease symptoms in patients with previously heavily treated relapsed or refractory HL and received FDA approval in 2016, followed by the approval of pembrolizumab in 2017 [105]. It is thought that compared to other hematologic malignancies, the benefit of PD-1 blockade is most clearly observed in HL patients.

RS cells are multinucleated large cells found in the lymph nodes of individuals with HL and used as a diagnostic measurement. Thus, RS cells have been thought to be a therapeutic target for treating HL. They are known as having B lymphocyte origin; however, the RS cells of classical HL fail to express most B cell functional genes and markers, such as CD20. RS cells are CD30-positive, and therefore, antibodies against CD30 are considered a therapeutic regimen. Brentuximab vedotin, an ADC that consists of an antibody against CD30 and a potent microtubule-disrupting agent, namely auristatins, was approved for treating HL and anaplastic large-cell lymphoma (ALCL) in 2011. Recently, trials on the combination of nivolumab and brentuximab vedotin in patients with relapsed or refractory classical HL showed a significant increase in complete remission rate compared with treatment with either nivolumab or brentuximab vedotin alone [106]. Brentuximab vedotin induced the depletion of CD30-positive RS cells, and the administration of nivolumab resulted in an increase of the T cell subset in peripheral blood [107].

### 3.4. Non-Hodgkin Lymphoma (NHL)

Non-Hodgkin lymphoma (NHL) is a cancer that begins in lymphocytes. While leukemia, such as acute lymphoblastic leukemia (ALL), mainly affects the bone marrow and blood, lymphomas affect the lymph nodes or other organs, possibly affecting the bone marrow as well. NHL is categorized based on the affected lymphocytes and the aggressiveness of the lymphoma [108]. Five types of NHL, namely diffuse large B cell lymphoma (DLBCL), follicular lymphoma (FL), small lymphocytic lymphoma (also known as chronic lymphocytic leukemia, CLL), marginal zone B cell lymphoma (MZL), and mantle cell lymphoma (MCL), occur more commonly. In particular, DLBCL is the most common type of aggressive lymphoma, whereas FL and CLL are slow-growing lymphomas [109]. As implied by the name, T cell lymphoma affects T cells, accounting for about 10% of all NHL cases.

#### Immunotherapy for NHL

In contrast to HL, PD-1 blockade has not resulted in obvious clinical responses in patients with NHL, including DLBCL and FL. Combination therapy with nivolumab and ibrutinib (a Bruton’s tyrosine kinase inhibitor), has also been assessed in relapsed or refractory DLBCL and FL, as well as CLL. However, the overall response was not different from that of ibrutinib monotherapy [110]. The tumor microenvironment of DLBCL has been described as immunologically “cold” based on transcriptional and histological studies, indicating low immunogenicity [111]. Of note, the tumor immune microenvironment is strongly associated with ICB efficacy, and there are some NHL subtypes exhibiting good response rates to PD-1 blockade. For instance, it was recently reported that patients with EBV-positive NHL responded to pembrolizumab more efficiently than those with EBV-negative NHL, presumably due to high PD-L1 expression in EBV-positive NHL [112]. Up to 25% of DLBCL patients harbor *PD*-*L1* gene alterations, resulting in PD-L1 overexpression. These cases represent low progression-free survival following front-line chemotherapy despite the high infiltration of clonal T cells. Thus, PD-1 blockade has resulted in good responses in these patients [113]. Importantly, a phase II study on the treatment of FL patients with the front-line use of a combination of nivolumab and rituximab resulted in an objective response rate (ORR) of 84% and CR of 47%, suggesting that the patients’ disease status might be important with respect to ICB response [114].

Exceptional successes of CD19 CAR-T cell therapies in clinical trials have led to achieving their FDA approvals. The first FDA-approved CAR-T, tisagenlecleucel, was treated for B cell ALL patients at 25 years of age or younger. Encouraging results have been reported with an ORR of 81% and relapse-free survival of 66% at 18 months [42]. In the following year, 2018, the same drug was approved for treating adults with relapsed or refractory DLBCL. Axicabtagene ciloleucel, another CD19 CAR-T cell, was also approved for relapsed or refractory large B cell lymphoma and DLBCL in 2017. A long-term follow-up of a phase I/II clinical trial using axicabtagene ciloleucel reported an ORR of 83% and a CR of 40% in RR DLBCL with the response persisting for over two years [115]. In 2020, brexucabtagene autoleucel was approved for treating mantle cell lymphoma. It consists of a similar CAR construct to axicabtagene ciloleucel, but the final CAR-T cell product has been improved by using a different autologous T cell enrichment process. It showed an ORR of 87% and a CR of 62% at six months [37].

Despite the overall long-term survival of up to 30% of treated patients, the remaining patients experience adverse events or relapse after CAR-T cell therapy. The major mechanisms for relapse are considered to be the poor persistence of CAR-T cells and loss of CD19 antigen expression on the malignant cells. The technical and biological obstacles for the production of autologous CAR-T cells also have to be overcome. Various novel strategies, including fourth-generation CAR-T cells, inducible CAR-T cells, and/or multiple antigen-targeting CAR-T cells, have been described for overcoming limited efficacy [116].

### 3.5. Multiple Myeloma (MM)

Multiple myeloma (MM) is the second most frequently diagnosed hematologic malignancy, and the most common type of myeloma [117]. It is characterized by a clonal expansion of aberrant plasma cells in the bone marrow, resulting in the production of an abnormal quantity of monoclonal immunoglobulins (Ig) called M protein. M proteins attack organs, such as kidney and bone, leading to end-organ damage. Since it remains an incurable malignancy, many therapeutic regimens, particularly immunotherapies using mAbs and CAR-T cell therapies, are emerging.

#### Immunotherapy for MM

Although an early clinical trial of combination therapy with pembrolizumab, lenalidomide, and low-dose dexamethasone reported good efficacy (ORR 44%) in relapsed and refractory MM patients [118], the following phase III trials revealed an unfavorable risk profile [119,120]. Recently, single-cell RNA-seq analysis of the MM tumor microenvironment indicated that a senescent and dysfunctional T cell subset is enriched in the bone marrow [121]. MM predominantly grows in the bone marrow, and the unique tumor microenvironment might prohibit reinvigoration of T cells in response to PD-1 blockade [122]. Furthermore, MM progression can trigger an increase in immunosuppressive subsets, such as myeloid-derived suppressor cells (MDSCs) and Treg cells. This led to a clinical trial of daratumumab (anti-CD38 mAb, against CD38^+^ Treg cells) in combination with an anti-PD-1 mAb for relapsed and refractory MM, which was terminated due to increased adverse events and less benefit [123].

In 2020, belantamab mafodotin (Blenrep) was approved as an ADC for the treatment of patients with relapsed or refractory MM who have received at least four prior therapies, including an anti-CD38 monoclonal antibody, a proteasome inhibitor, and an immunomodulatory agent. The drug is a monoclonal antibody specific to B cell maturation antigen (BCMA) and linked to toxic drug auristatin F. This ADC attracted attention as it was the first approved immunotherapy targeting BCMA. BCMA is an important target for treating MM, and BiTEs targeting it are also being developed.

Several CAR-T cell therapies are being investigated for MM, and the most advanced one targets BCMA. BCMA CAR-T therapy had outstanding results in heavily pretreated MM patients [124], and it was filed for US FDA approval in March 2020. The overall response rate of BCMA CAR-T cell therapy was 73.4% with 31.3% of patients achieving a complete response. SLAMF7 is a member of the signaling lymphocytic activation family of receptors and regulates the immune system. It is enriched on malignant plasma cells’ surface, while also expressed on other immune cells, including NK cells, T cells, B cells, and macrophages, but not on HSCs or nonhematopoietic cells [125]. Notably, an anti-SLAMF7 mAb (elotuzumab) in combination with lenalidomide exhibited a response without significant adverse events in MM, leading to the development of SLAMF7 CAR-T cells [126]. Additionally, CD44v6, an isoform of the hyaluronate receptor, is also targeted by CAR-T therapy for treating patients with MM. Remarkable antimyeloma effects were reported for CD44v6 CAR-T cells in a mouse model [127,128].

A summary of types of hematologic malignancies, incidence rates, and immunotherapies that received FDA approval and described in Section 2 and Section 3 is shown in Figure 1.

## 4. Novel Targets and Emerging Therapeutic Approaches

### 4.1. Emerging Targets for Immune Checkpoints

#### 4.1.1. Immune Checkpoint Receptors on T/NK Cells

ICB predominantly depends on T cells for therapeutic efficacy. However, harnessing NK cells or macrophages through ICIs may be a potential novel approach. NK cells can eliminate transformed cells that lose MHC-I expression (“missing-self recognition”) or express danger ligands (“induced-self recognition”) [129]. This response can be modulated by various activation and inhibitory receptors expressed on mature NK cells, such as NKG2D, DNAM-1, or NKG2A and KIR, respectively. Activation receptors recognize ligands upregulated on tumor cells, while inhibitory receptors recognize HLA-E [130]. ICIs that enhance NK cell activation could target inhibitory receptors through an anti-NKG2A mAb or anti-KIR mAb. NKG2A/CD94 dimers recognize ubiquitously expressed nonclassical HLA-E, and HLA-E molecules can be overexpressed by certain tumors [131]. It was demonstrated that an anti-NKG2A/CD94 mAb (monalizumab) in combination with cetuximab (anti-EGFR mAb) had therapeutic efficacy in patients with head and neck cancer [132]. In addition, anti-KIR mAb (IPH2102) in combination with lenalidomide resulted in clinical benefit in relapse and refractory MM patients [133].

Lymphocyte-activation gene 3 (LAG-3, CD223) and T cell immunoglobulin and mucin domain 3 (TIM-3) are coexpressed with PD-1 on tumor-infiltrating lymphocytes [134,135]. Upon the binding of LAG-3 and TIM-3 to their ligands, fibrinogen-like protein 1 (FGL1) and galectin-9, respectively, CD8^+^ T cells are impaired and cannot execute cell-mediated antitumor immunity functions [136]. Thus, LAG-3 and TIM-3 are considered T cell exhaustion markers similar to PD-1. FGL1 is often upregulated in solid tumor tissues, and, as of recently, clinical trials are being conducted to test the safety and efficacy of targeting LAG-3 in combination therapy with PD-1 (NCT03005782 and NCT02061761) as well as to assess a BiTE against PD-1 and LAG-3 (NCT03219268). TIM-3 and its ligand galectin-9 are widely expressed in various types of cancer, including hematologic malignancies [137]. In particular, TIM-3 is an LSC marker, and galectin-9 promotes the self-renewal of transformed cells [138]. Therefore, TIM-3 blockade could have therapeutic potential in AML.

T cell immunoreceptor with Ig and ITIM domains (TIGIT) is present on T cells and NK cells and is an inhibitory counterpart of DNAM-1 [139]. TIGIT recognizes CD155 as a ligand, which can also interact with DNAM-1, and transmits inhibitory signaling, resulting in the negative regulation of T cell- or NK cell-mediated antitumor effects [140]. In AML patients, the DNAM-1^low^PD-1^+^TIGIT^+^ T cell subset is observed, and an increase in this subset correlates with poor prognosis [141]. CD155 is widely expressed in malignancies, including blood cancers. Additionally, TIGIT is the most frequently expressed checkpoint molecule in MM [142]. Taken together, the blockade of TIGIT might represent a promising immunotherapy approach against hematologic malignancies.

CD70, a member of the tumor necrosis factor receptor ligand family, is highly expressed in various solid tumors and hematologic malignancies but has limited expression in normal tissue. It interacts with its receptor CD27, which is constitutively expressed on naïve T cells, NK cells, and HSCs, while being downregulated on effector T cells [143]. Normally, signaling through CD27 is implicated in the differentiation, activation, and survival of lymphocytes [144]. However, CD27 has also been shown to promote LSC growth and disease progression in leukemia [145]. Besides the current investigation for AML as mentioned in Section 3.1., anti-CD70 mAb, has also been investigated in heavily pretreated patients with CD70-expressing advanced cutaneous T cell lymphoma. Recently, ADCs, including anti-CD70 mAb or anti-CD27 mAb with azacitidine, have been investigated for treating hematologic malignancies [146].

#### 4.1.2. Immune Checkpoint Receptors on Tumor-Associated Macrophages

Macrophages also are novel candidates for immune checkpoint blockade through antibody-dependent cellular phagocytosis (ADCP) [147]. CD47, which, as previously mentioned, acts as a “don’t eat me” signal, is a ligand of signal regulatory protein alpha (SIRPα) on macrophages, and SIRP alpha receptor signaling attenuates their phagocytic activity upon interaction with its ligand [148]. CD47 overexpression is often observed in malignant tumor cells, and CD47 blockade was reported to be effective for lymphoma cell clearance [149]. A combination of anti-CD47 mAb (Hu5F9-G4) and rituximab resulted in a good response in clinical studies, with a CR of 43% in FL patients and 33% in DLBCL patients [150]. In 2018, a second “don’t eat me” signal was identified, namely the leukocyte immunoglobulin-like receptors subfamily B (LILRBs), which recognize MHC class I molecules on tumor cells and induce resistance to anti-CD47 mAb therapy [151]. LILRBs are widely expressed, including on immune cells, and have been investigated as novel immunotherapy targets [152]. For instance, it has been suggested that LILRB4 CAR-T cells are effective for targeting LILRB4^+^ AML, as LILRB4 is limitedly expressed on monocytes and monocytic AML cells [153]. Other “don’t eat me” signals, including Siglec-10, CD24, and the adenosine 2A receptor (A2AR), have also been identified [154,155].

#### 4.1.3. Novel Candidates for Immune Checkpoint Blockade

New immune checkpoint pathways are constantly being uncovered. More immune checkpoint targets remain to be thoroughly investigated, and four of them are briefly introduced herein: (1) Among the IgG Fc receptor (Fc gamma receptor) family, CD32A is restrictively expressed in LSCs but not in healthy HSCs, suggesting that it could be a target for immunotherapy [156]. (2) CD96 is a receptor for CD155 and plays a role in the interaction between NK cells and target cells. Interestingly, CD96 is upregulated in AML LSCs with increased relapse rate and poor chemotherapy responses, indicating that an anti-CD96 antibody could be used for targeted immunotherapy [157]. (3) CD105 (endoglin) is expressed on endothelial cells and part of the TGFβ receptor complex, participating in tumor-associated angiogenesis. CD105 expression on malignant blasts enhances leukemogenic activity, and an anti-CD105 mAb reduced AML engraftment in an animal model [158]. (4) CD200 is an inhibitory ligand of the immunoglobulin superfamily and is expressed in the brain, testis, and hematopoietic cells [159]. It is also overexpressed in AML blasts, and, interestingly, an anti-CD200 mAb (samalizumab) is currently tested in a clinical trial (NCT03013998).

### 4.2. Novel Approaches for CAR-T Cell Therapy

Similarly to CD19, CD22 is expressed on most malignant B cells. Thus, it is proposed as an alternative CAR target for treating B-ALL and relapsed disease following CD19 CAR-T cell therapy [160]. Since relapse after CD19 CAR-T cell therapy partially occurs due to loss of CD19 antigen expression on malignant cells, CAR-T with a novel target or dual CAR targeting both CD19 and CD22 may overcome the resistance. There are several ongoing clinical trials of CAR-T cells against CD22.

Other possibilities for combination treatments involving CAR-T cells against multiple antigens are presented with the introduction of BCMA CAR-T for treating MM. CD138 is highly expressed in MM as well as other tissues, and CD138 CAR-T cells only achieved modest responses [161]. This can be overcome through a combination of BCMA CAR-T cells. Similarly, CD38 is overexpressed in MM cells and is also observed in other cells. Interestingly, CD38 CAR-T cells were modulated to have a low affinity to discriminate between CD38 high tumor cells and CD38 low normal cells through the redesign of CAR using light-chain exchange technology [162]. G protein-coupled receptor class C group 5 member D (GPRC5D) CAR-T [163] and CD56 CAR-T cells have also been developed and proposed for combination therapy with BCMA CAR-T [164].

The cancer-testis antigen NY-ESO-1 is expressed in a variety of malignant neoplasms. MM patients exhibited NY-ESO-1 expression in 60% of cases at diagnosis but in 100% of cases at relapse with poor prognosis [165]. Of note, NY-ESO-1 is an intracellular antigen and is recognized only by TCRs. CAR-T cells have been generated to recognize the NY-ESO-1/HLA complex on the tumor cell surface, and these CAR-T cells were effective [166]. A phase I/II clinical trial has been registered to evaluate the safety and efficacy of NY-ESO-1 CAR-T cells in MM. Importantly, CAR-T cell therapies for hematologic malignancies have been diversified and advanced, not only by investigating novel antigen targets, but also by modifying CAR designs, improving protocols for gene transduction and cell expansion, as well as inactivation of the TCR α constant (*TRAC*) gene [167,168]. The next goal in CAR-T/CAR-NK cell therapies is to generate universal CAR-T cells that can provide improved therapeutic results in more malignancies in addition to B-ALL and DLBCL.

Various targets for novel immunotherapies in hematologic malignancies, including tumor-associated antigens and immune checkpoint receptors, are depicted in Figure 2.

### 4.3. Cancer Vaccinations

As mentioned above, WT1 has been used to stimulate the patient’s immune system by targeting disease-specific cells via priming immune cells against specific tumor antigens for AML. A phase I/II trial with the WT1 peptide vaccine revealed that WT1 specific cytotoxic T cells were efficiently proliferated, resulting in a reduction of leukemic blasts [169,170]. Proteinase 3 has also been studied as a targeted antigen for vaccination in AML as it is a serine protease that is overexpressed in AML cells [171]. DC vaccines derived from leukemia-associated antigens have also been proposed; however, they seem to induce weak responses, presumably due to tolerance.

## 5. Conclusions

We believe that immunotherapies will constitute a common treatment regimen for certain types of hematologic malignancies in the near future. This is due to the exceptional clinical outcomes with complete remission achieved through approaches such as CAR-T cells for B-ALL and DLBCL. However, the response can be affected by various factors, and immunotherapy regimens may not be suitable for certain patients, thus highlighting the need for biomarkers to determine the best treatment approach [172]. Another successful example is PD-1 ICIs, for which the need for biomarkers of treatment efficacy has already been discussed in depth. In addition, hematologic malignancies are more frequent in elderly people, and many countries are now experiencing aging societies. Thus, novel immunotherapies that can boost patients’ immunity or gene and cell therapies that can be used in allogeneic settings such as universal CAR-T cells will be required. In addition, mAbs can be potentially implicated to develop novel immunotherapies including BiTEs, TriKEs, CAR-T, CAR-NK, and ADCs, as well as in combination with chemotherapy. Taken together, more immunotherapy regimens will be employed for the treatment of hematologic malignancies in the future, and we expect that more precise, efficient, and safer therapies will be developed.

## Figures and Tables

**Figure 1 ijms-21-08000-f001:**
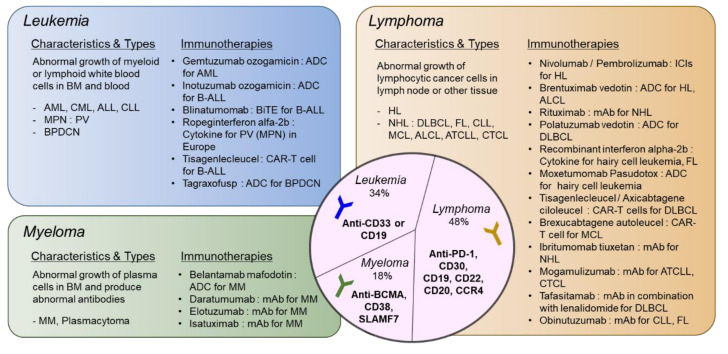
Types of hematologic malignancies and immunotherapies that received FDA approval. Hematologic malignancies are categorized into leukemia, lymphoma, and myeloma. Disease characteristics and subtypes are described and currently available immunotherapies approved by FDA are listed. In the middle circle, targets for immunotherapeutic approaches specific to each disease type are shown. ADC: antibody–drug Conjugate; ALCL: anaplastic large-cell lymphoma; ALL: acute lymphoblastic leukemia; AML: acute myeloid leukemia; ATCLL: acute T cell leukemia/lymphoma; B-ALL: B cell acute lymphoblastic leukemia; BPDCN: blastic plasmacytoid dendritic cell neoplasm; CAR: chimeric antigen receptor; CLL: chronic lymphocytic leukemia; CML: chronic myeloid leukemia; CTCL: cutaneous T cell lymphoma; DLBCL: diffuse large B cell lymphoma; FL: follicular lymphoma; HL: hodgkin lymphoma; ICI: immune checkpoint inhibitor; MCL: mantle cell lymphoma; MM: multiple myeloma; MPN: myeloproliferative neoplasm; NHL: non-Hodgkin lymphoma; PV: polycythemia vera. please refer to the manuscript for other abbreviations. The list of drugs can be searched at https://www.cancer.gov/about-cancer/treatment/drugs/.

**Figure 2 ijms-21-08000-f002:**
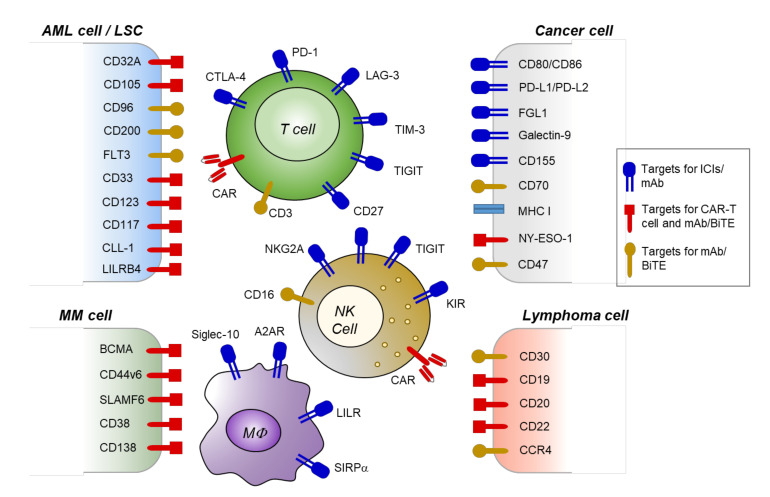
Emerging immunotherapies with new targets for hematologic malignancies. Targets for immunotherapies that are effective in preclinical or clinical trials are depicted. Blue receptors/antigens are immune checkpoints. Molecules in red represent targets for monoclonal antibody (mAb), bispecific T cell engager (BiTE), or chimeric antigen receptor (CAR)-T/NK therapies, while dark yellow means immunotherapy targets that have not been investigated for the CAR-T cell approach.

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
