# Peer review of "Immunotherapy in Hematologic Malignancies: Emerging Therapies and Novel Approaches"

_ijms, 2020, doi:10.3390/ijms21218000_

Round 1

Reviewer 1 Report

Noh and colleagues present a work that describe immunotherapies in hema malignancies.

The work is in a good English, and quite easy to read. Introduction is nice and concise. Conclusion is too enthusiastic, and I think that nowadays it is impossible to state that immunotherapy will replace target therapy or chemo.

The figures presented are very good-looking, some tables could be added to improve clarity of very long sections (e.g. 3.5.1 and 4.1)

I've 3 major observations:

  • the work is not well balanced, considering the development of immunotherapies in different fields of hematology. The section on acute myeloid takes a large part of the work, even if immuno-therapy approved for AML consists only in an ADC. On the counter part, there is not a specific section on ALL and HL received a strict space.
  • Introductions on some diseases (e.g. introduction on AML/MDS (3.1) MPN (3.2)) are too extensive and present general data on the diseases; I advice to focus on disease features that justify the use of immuno-therapies. This will better fit with the scope of this review.
  • Please pay attention in not fragmentating information, putting together all the info needed for the understanding of a malignancy/setting. 

Minor revision are needed:

  • abstract: NGS is never treated in the work in relation to immunotherapy
  • introduction, page 1 line 35: prefer drug name to brand name, if it is possible.
  • introduction, page 1 line 37: the sentence is too enthusiastic
  • page 2, line 49-51: maybe some lines remained from the submission format
  • 2.1 page 2 line 60-63: IL-2 is also used in the context of adoptive NK cell therapy in AML, e.g. Curti et al
  • page 2 line 54 and subsequent: INF, please specify interferon type (y??)
  • 2.4 page 3 line 117-119: this sentence is too pessimistic; BiTe showed very good results in ALL and blinatumomab is approved, however these results could be improved.
  • 2.5 page 3 line 135: nejm on car NK cell also showed safety of antiCD19 CAR NK
  • 2.5 page 3 line 142: other than domains also linkers have been showed to influence CART cell activity
  • 2.5 page 4 line 144: in our opinion, will limits of autologous CAR T be passed by allogeneic CAR T, off-the-shelf CAR T and HLA- CAR NK?
  • 3 page 4 line 157-165: the greatest part of this introduction refer to AML, not to hema malignancies
  • 3.1 page 4 line 170-171: the sentence is not clear, please revise
  • 3.1 page 4 line 175: NPM1 is missing, please whenever report gene in a malignancy report by incidence.
  • 3.1 page 5 line 194: IDO and tryptophan metabolism, mesenchymal stem cell function are pivotal in AML immunology
  • 3.1/ 3.1.1 HLA loss is one of the better described mechanisms of relapse in AML (vago et. al)
  • page 6 line 252: is the paragraph break correct?
  • 3.1.1: does CD70 deserve to be treated in AML paragraph?
  • 3.2.1 page 6 line 281: please revise the sentence, it is not clear
  • 3.3 page 7 line 298-...: most of HL are sporadic, please revise this section accordingly
  • 3.3.1 page 7 line 316: please revise the sentence, someone could get that brentuximab contains both maytansinoids and auristatins (2 molecules)
  • page 8 line 344: CAR-T cell do not rely on tumor micro-environment as much as other drugs
  • Figure 1: please consider to remove etc. and replace with actual name of the most important other malignancies
  • Please consider to add tagraxofusp at the work, as it is FDA approved and very important even if used in a very rare disease
  • please cosnider to update drug code with actual drug name (e.g. argx-110 --> cusatuzumab) or to brovide both name and code.
  • Conclusion: please reflect on the possibility to use immunotherapy alone, that could be considered too enthusiastic with actual data for most of the diseases. In most of the hema malignancies target therapies gives similar or superior benefits if compared with immuno-threapy. Furthermore extensive works exist on the immunomodulatory role of chemotherapy (kroemer, galluzzi), please consider the potential implications in term of efficacy on immuno-therapy alone vs chemotherapy+immuno-therapy.

Author Response

We appreciate the reviewer’s comprehensive comments throughout the manuscript.

Please check point-by-point response file.

Reviewer 2 Report

Authors present a quality and well-written manuscript review that reviews emerging therapies and novel approaches in the immunotherapy of hematological malignancies.

Authors overview various important players in immunotherapy, including cytokines, immune checkpoint blockade (ICB), antibody-drug conjugates (ADCs), bispecific T/NK-cell engagers (BiTEs/BiKE), chimeric antigen receptor (CAR)-T, CAR-NK cells.

Authors cover a wide range of cancers, including AML/MDS, NHL, MM. They also discuss emerging targets for immune checkpoints and novel approaches for CAR-T cell therapy.

Comments:

- Line 54. Sign “gamma” is missing for INF-gamma. Line 122 - same problem.

- Line 56. Sign “beta“ is missing for IL-2 receptor beta.

- For part 2.5. Please also mention a third approved CAR-T therapy Thecartus. Add brief information about anti-CD20 or dual anti-CD19/CD20 CAR-T for B cell malignancies.

- Line 420. Sign “alpha” is missing for SIRP-alpha.

- Line 467. Sign “gamma” is missing?

- Authors might consider adding 1-2 more nice figures to improve the visual perception of their article.

- Authors are highly encouraged to cite the following article that overviews the challenges of applying CAR-T cell therapy for treatment of tumors.  https://doi.org/10.3390/cancers12010125

Overall, the manuscript is valuable for the immunotherapy scientific community and should be accepted for publication.

Author Response

(The authors gave the same response as above.)

Round 2

Reviewer 1 Report

Comments were addressed